# PROCEEDINGS A

# Research

crystal engineering, chemical engineering

calcium carbonate, crystal habit, Ostwald rule of stages, polymorphism, surface energy

**Author for correspondence:**
Eftychios Hadjittofis
e-mail: fc4eh@sheffield.ac.uk

# The role of surface energy in the apparent solubility of two different calcite crystal habits

Eftychios Hadjittofis[1,2], Silvia M. Vargas[3], James D. Litster[1] and Kyra L. Sedransk Campbell[1]

[1]Department of Chemical and Biological Engineering, The University of Sheffield, Mappin Street, Sheffield S1 3JD, UK
[2]UCB Pharma SA, Chemin du Foriest, B-1420 Braine-l'Alleud, Belgium
[3]BP America, Inc., Houston, TX 77079, USA

EH, 0000-0001-8107-8368

The interplay between polymorphism and facet-specific surface energy on the dissolution of crystals is examined in this work. It is shown that, using cationic additives, it is possible to produce star-shaped calcite crystals at very high supersaturations. In crystallization processes following the Ostwald rule of stages these star-shaped crystals appear to have higher solubility than both their rhombohedral counterparts and needle-shaped aragonite crystals. The vapour pressures of vaterite, aragonite, star-shaped calcite and rhombohedral calcite crystals are measured using thermogravimetric analysis and the corresponding enthalpies of melting are obtained. Using inverse gas chromatography, the surface energy of the aforementioned crystals is measured as well and the surface energy of the main crystal facets is calculated. Combining the effect of facet-specific surface energies and the enthalpies of melting on a modified version of the classical solubility equation for regular solutions, it is proved that the star-shaped calcite crystals can indeed have higher apparent solubility than aragonite crystals.

## 1. Introduction

In his work 'On the equilibrium of heterogeneous substances' [1], Prof. J. W. Gibbs discusses the role of

interfacial tension (denoted by $\sigma$) in dissolution, stating that:

> With these preliminary notions, we now proceed to discuss the condition of equilibrium which relates to the dissolving of a solid at the surface where it meets a fluid, when the thermal and mechanical conditions of equilibrium are satisfied. It will be necessary for us to consider the case of isotropic and of crystallized bodies separately, since in the former the value of $\sigma$ is independent of the direction of the surface, except so far as it may be influenced by the state of strain of the solid, while in the latter the value of $\sigma$ varies greatly with the direction of the surface with respect to the axes of crystallization, and in such a manner as to have a large number of sharply defined minima.

This statement highlights the role of surface energy anisotropy of crystalline materials in dissolution. Following its proposal by the pioneer of thermodynamics, the role of surface energy has been applied particularly in the area of mineralogy [2]. The lack, for a long period, of a methodology to determine the surface energy anisotropy of samples of particles limited the advancement of this field and did not allow its expansion. In particular, it did not allow the direct incorporation of experimentally determined surface energy values. However, researchers from different disciplines have approached this topic from alternative pathways [3–7].

Incorporating the concept of surface energy anisotropy, i.e. that crystals are faceted particles with each facet carrying different surface energy (depending on the facet-specific surface chemistry), in the context of the regular solution theory, the following equation can be derived [8]:

$$\ln(X) = \ln(X^{\text{ideal}} + X^{\text{surface}}) = \int_{T_m}^{T} \frac{-\Delta H_m(T)}{T^2} dT + \frac{1}{T} \sum_{i=1}^{n} (A_i (W_{il}))$$

$$= \int_{T_m}^{T} \frac{-\Delta H_m(T)}{T^2} dT + \frac{1}{T} \sum_{i=1}^{n} \left( A_i \left( 2\sqrt{W_{cc,i} W_{ll}} \right) \right). \tag{1.1}$$

In equation (1.1), $X$ is the mole fraction of the solute in solution, $X^{\text{ideal}}$ is the mole fraction based on the ideal solution theory and $X^{\text{surface}}$ stands for the contribution of surface energy in the ideal solution component, $\Delta H_m$ is the enthalpy of melting of the compound of interest, $T$ and $T_m$ are the temperature and melting temperature, respectively, $A_i$ is the surface area of facet $i$ per mole, $W_{cc,i}$ is the work (in $J\,m^{-2}$) required to remove a solute molecule from facet $i$ of crystal $c$, $W_{ll}$ is the work required to remove a bulk liquid molecule to effectively create space for a solute molecule and $W_{il}$ is the work required for a solute molecule from facet $i$ to get into the bulk liquid. Therefore, the terms $W_{ii}$ and $W_{il}$ are facet-dependent quantities and they are proportional to the facet-specific surface energies. The first part of the right-hand side of the equation comes from the ideal solution theory, whereas the second part includes the effects of surface energy anisotropy.

During crystallization at high supersaturation, obeying the Ostwald rule of stages [9–13], the formation of the thermodynamically most stable crystalline form is the outcome of a sequential process, where the pathway comprises crystallization and dissolution of metastable forms. Based on equation (1.1) it can be suggested that the term form, in the above statement, not only refers to the different polymorphs, each one with different enthalpy of melting, but also expands to different crystal habits, each one exhibiting different crystal facets, at different ratios. Nevertheless, the majority of studies on the effects of the Ostwald rule of stages on crystallization describe solely polymorphic transformations.

This work uses $CaCO_3$ as the model compound to demonstrate the interplay between polymorphism and surface energy in the Ostwald rule of stages. As the most abundant mineral on Earth, $CaCO_3$ has been thoroughly investigated and found to have three polymorphs: vaterite, aragonite and calcite (from the least to the most stable) [14–17]. Previous experimental studies at sufficiently high supersaturation showed that the crystallization proceeds following the Ostwald rule of stages: snowflake-shaped vaterite crystals emerge first; these then dissolve and are substituted by needle-shaped aragonite crystals, which in turn dissolve to be substituted by rhombohedral calcite crystals [18]. In this work, the crystallization occurred in sweet and anoxic

conditions in the presence of $Ni^{2+}$ ions. Under these conditions vaterite crystals emerge first, followed by star-shaped calcite crystals. The star-shaped calcite crystals then dissolve and needle-shaped aragonite crystals are formed, which in turn dissolve to be substituted by rhombohedral calcite crystals.

The vapour pressures of vaterite, aragonite and both the rhombohedral and star-shaped calcite were measured using thermogravimetric analysis (TGA). The results show that vaterite has the highest vapour pressure, followed by aragonite. The two types of calcite have similar vapour pressures, suggesting that they are thermodynamically identical. The surface energy was measured by means of inverse gas chromatography (IGC). The results show that the star-shaped calcite crystals have much higher surface energy ($\gamma^{total}$) than their rhombohedral counterparts and aragonite. Analysis of the different components of the surface energy—the van der Waals forces, the acid and the base—indicates that the star-shaped crystals have better wettability with water, owing to the higher surface energy of their facet.

In this paper, the vapour pressure data for all three polymorphs of $CaCO_3$ are measured for the first time, as well as the corresponding surface energies. The crystal habit modification and the consequential Ostwald rule of stages phenomenon, achieved in the presence of $Ni^{2+}$, are very didactic with respect to the interplay between polymorphism and surface energy as they demonstrate that the surface energy anisotropy can be a crucial parameter in the determination of solubility.

## 2. Methods and materials

### (a) Synthesis of CaCO₃

Deionized water at 80°C was de-aerated for 3 h under a stream of $N_2$ (200 ml min$^{-1}$, 1 bar) in a 250 ml round bottom flask. This work is part of a greater effort to investigate phenomena under sweet and anoxic conditions. Such conditions are encountered industrially in the oil and gas industry, but they are also relevant to other scientific disciplines. In particular, disciplines investigating phenomena in Earth's primordial period often consider sweet and anoxic conditions. Therefore, it was important to remove all the oxygen to ensure consistency with the broader context of this work. De-aeration under these conditions ensures that all the oxygen is removed from the flask [19]. Following de-aeration, the gas stream was switched to $CO_2$ (200 ml min$^{-1}$, 1 bar) for 12 h. Supersaturation ($S_{CaCO3}$), with respect to $CaCO_3$, was achieved with the addition of $CaCl_2 \cdot 2H_2O$ (Sigma Aldrich), which was dissolved in 10 ml of de-aerated deionized water. Under pristine conditions, i.e. in the absence of any additives, this amount of $CaCl_2 \cdot 2H_2O$ corresponds to a $S_{CaCO3}$ of ca 30 000 [8,9]. When required, the pH of the solution was adjusted to ca 7.2 using $NaHCO_3$ (Sigma Aldrich). Experiments were continuously purged with $CO_2$ (200 ml min$^{-1}$, 1 bar) for different durations: 3 min, 5 min, 24 h, 48 h and 240 h. After each experiment, the solution was filtered to remove particles smaller than 3 µm from the solution. The harvested solid was analysed. Replicates of experiments were performed to ensure that sufficient product yield could be obtained in all cases. Powders were stored under a dry $N_2$ environment individually, but combined for analysis.

### (b) Synthesis of CaCO₃ in the presence of Ni²⁺

The above synthetic procedure was modified with the introduction of $Ni^{2+}$ ions, as additives, in the form of $NiCl_2 \cdot 6H_2O$ (Sigma Aldrich). Thus, 0.4 g of $NiCl_2 \cdot 6H_2O$ was dissolved along with the $CaCl_2 \cdot 2H_2O$ in 10 ml of de-aerated deionized water and then injected into the flask.

### (c) Synthesis of CaCO₃ polymorphs

Vaterite, aragonite and rhombohedral calcite were synthesized in this work. For the synthesis of vaterite, 0.5 g of $CaCl_2 \cdot 2H_2O$ was injected into the round bottom flask, when purging with

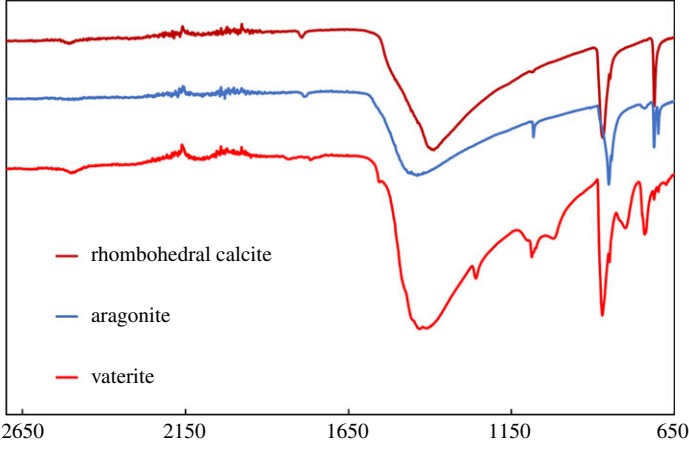

**Figure 1.** The FTIR spectra obtained for vaterite, aragonite and rhombohedral calcite. (Online version in colour.)

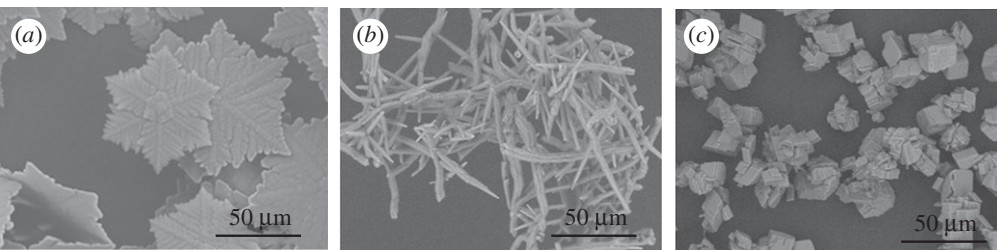

**Figure 2.** SEM images of (*a*) vaterite obtained from 0.5 g of $CaCl_2 \cdot 2H_2O$ crystallized for 10 min, (*b*) aragonite obtained from 0.5 g of $CaCl_2 \cdot 2H_2O$ crystallized for 2 h and (*c*) rhombohedral calcite obtained from 2 g of $CaCl_2 \cdot 2H_2O$ crystallized for 96 h. All the experiments were conducted under the conditions described in §2c.

$CO_2$ was complete, and left to crystallize for a duration of 10 min. For the synthesis of aragonite, 0.5 g of $CaCl_2 \cdot 2H_2O$ was injected into the flask and allowed to crystallize for 2 h. For the synthesis of vaterite, 2 g of $CaCl_2 \cdot 2H_2O$ was injected into the flask and allowed to crystallize for 96 h. In figure 1 the corresponding Fourier transform infrared (FTIR) spectra are shown, and in figure 2 scanning electron microscope (SEM) images of the crystals obtained from the different polymorphs are shown. Vaterite has a snowflake crystal habit, aragonite crystals are needle shaped and calcite crystals are rhombohedral.

## (d) Characterization of CaCO₃ crystals

The dried solid products were assessed via X-ray diffraction (XRD); an X'Pert PRO diffractometer (PANalytical) was used. The XRD measurements were conducted at a $2\theta$ range from 5° to 80°, with a step size of 0.01° $(2\theta)$ and a count time of 1 s. A back-loaded sample holder was used. As the material was relatively free flowing and fine, no mechanical force was exerted on it to assist in the packing. Owing to the fine nature of the particles, the surface of the sample can be considered flat with respect to the sample holder. For the crystals produced at 3 and 5 min, in the presence of $Ni^{2+}$, the material was unloaded from the sample holder, mixed with the mother batch and then a new sample was taken for a second measurement. The same procedure was repeated once more. The three measurements were identical.

A Hitachi TM-100 table-top SEM (Hitachi Ltd, Tokyo, Japan) and a JSM-6010LA (Jeol Ltd, Tokyo, Japan) were used to image the materials produced.

The TGA was conducted using a Netzsch STA449 (Netzsch-Gerätebau GmbH, Selb, Germany). The $CaCO_3$ samples were placed in alumina pans (crucibles) and pre-treated *in situ* at 200°C, prior to the actual measurement, to remove unbound moisture.

A methodology similar to the one found in the literature was used for the determination of vapour pressure of the $CaCO_3$ samples [20,21]. The foundation of this methodology is the following equation for the rate of mass loss ($dm/dt$) per unit area ($\alpha$) during the heating of a compound:

$$\left(\frac{1}{\alpha}\right)\frac{dm}{dt} = Pa\sqrt{\frac{M}{2\pi RT}}. \tag{2.1}$$

In the above equation, $P$ is the vapour pressure, $a$ is a vaporization coefficient, $M$ is the molar mass of the vapour (in the case of $CaCO_3$ the vapour is $CO_2$) and $R$ and $T$ are the universal gas constant and temperature, respectively. The value of $\alpha$ is determined by the diameter of the orifice, which in this case is assumed to be equal to the diameter of the TGA's pan. The vaporization constant was determined using benzoic acid as the calibration standard. Benzoic acid has a very well reported vapour pressure curve. Thus, it can be used to determine the value of the vaporization coefficient at the particular arrangement. The vapour pressure measurements were conducted from 500°C to 620°C.

The surface energy measurements were performed using an IGC-Surface Energy Analyser (Surface Measurement Systems, London, UK). The experiments were conducted at 30°C under a helium flow rate of 10 sccm. The dispersive component of the surface energy ($\gamma^{LW}$) was determined using a series of chain alkanes (heptane, octane, nonane and decane) and the acid–base component ($\gamma^{AB}$) was measured using monopolar probes (toluene, ethyl acetate, acetonitrile and dichloromethane) [22–25]. The centre of mass of the chromatograms was used in data analysis and the Della Volpe method was used for the determination of $\gamma^{AB}$. The Schultz method was used for the determination of $\gamma^{LW}$ and the presented results have an $R^2 > 0.999$ [26]. The measurements were performed at target surface coverages ranging from 0.01 to 0.15. The isotherms for the materials under investigation were determined using octane; $0.09\,m^2\,g^{-1}$ for vaterite, $0.9\,m^2\,g^{-1}$ for aragonite, $0.6\,m^2\,g^{-1}$ for star-shaped calcite and $0.2\,m^2\,g^{-1}$ for the rhombohedral calcite. The total surface energy of a material ($\gamma^{total}$) is given as follows:

$$\gamma^{total} = \gamma^{LW} + \gamma^{AB} = \gamma^{LW} + 2\sqrt{\gamma^+\gamma^-}. \tag{2.2}$$

In equation (2.2) $\gamma^+$ and $\gamma^-$ stand for the acid and base components of the surface energy, respectively.

Surface energy measurements were conducted on aragonite, rhombohedral calcite and star-shaped calcite samples. The work of adhesion between the material and water ($W_{AB}$) was calculated based on these measurements using the equation:

$$W_{AB} = 2\left(\sqrt{\gamma_{solid}^{LW}\gamma_{water}^{LW}} + \sqrt{\gamma_{solid}^+\gamma_{water}^-} + \sqrt{\gamma_{solid}^-\gamma_{water}^+}\right). \tag{2.3}$$

The magnitude of $W_{AB}$ provides a metric for the affinity of the material with water; the higher the magnitude, the more hydrophilic the surface. Similarly to the IGC measurements, the Della Volpe values for the $\gamma^+$ and $\gamma^-$ of water were used in the calculations [27].

The surface energy distributions of the surface energy maps produced by the IGC measurements were deconvoluted using *in silico* tools described in the literature in order to identify the surface energy of the main crystal facets [28,29]. The surface energy plot obtained by an IGC measurement is determined by the adsorption of the different solvent probes on the different crystal facets. Each crystal facet, of each polyomorph, carries a different surface energy that appears in different proportions depending on the crystal habit. This *in silico* methodology simulates the adsorption process taking place during an IGC measurement. It starts with some initial guesses for the number of facets with distinct values of surface energy, the magnitude of the surface energy of each facet and the relative abundance of each facet. It then uses an optimization algorithm to identify the best combination of these parameters to fit the experimentally obtained IGC plot.

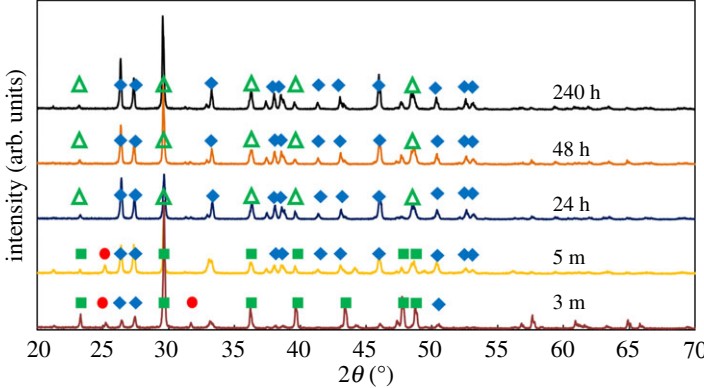

**Figure 3.** The XRD patterns of the powders obtained from the crystallization of 6 g of $CaCl_2 \cdot 2H_2O$ in the presence of 0.4 g of $NiCl_2 \cdot 6H_2O$. The crystallization time is given next to each pattern. The red circles indicate the characteristic peaks of vaterite, the blue rhombi are for the characteristic peaks of aragonite, the green rectangles are for the cases where the SEM images suggest the presence of star-shaped calcite crystal and the open green triangles represent the peaks of calcite for the cases where SEM images suggest the coexistence of star-shaped and cuboid crystals. (Online version in colour.)

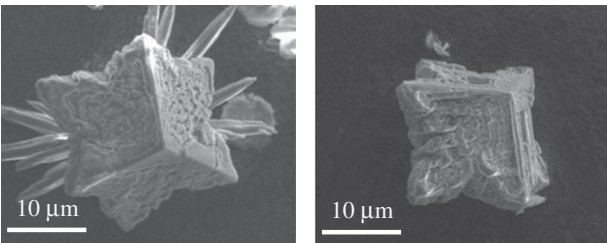

**Figure 4.** Star-shaped calcite crystals obtained from the crystallization of 6 g of $CaCl_2 \cdot 2H_2O$ in the presence of 0.4 g of $NiCl_2 \cdot 6H_2O$ for 3 min. (Online version in colour.)

## 3. Results

Figure 3 shows the XRD patterns for material obtained from crystallization experiments as described in §2b, at different time periods. Analysis of SEM images suggests that the material obtained for experiments run for less than 24 h contains star-shaped calcite crystals, such as those reported previously in the literature. Star-shaped crystals obtained from the experiments conducted for 3 min are shown in figure 4. The material obtained after 24 h contains a mixture of these crystals with the well-known rhombohedral calcite crystals. Vaterite has been observed at the onset of crystallization as well. However, aragonite is the dominant polymorph in all the experiments run for more than 5 min.

Calcite has its main characteristic XRD peak at approximately 29.55° (peak 1), whereas aragonite has two characteristic peaks at approximately 26.25° (peak 2) and approximately 27.25° (peak 3). The intensities of these peaks for each case are shown in table 1. In the same table, the ratios between the three peaks are shown. The results suggest a decrease in the ratio of the calcite peak to the aragonite peaks up to 24 h. Then there is a shift and the calcite peak grows compared with the other aragonite peaks. Similarly, it is interesting that even though traditionally the 26.25° peak of aragonite is higher than the 27.25° one, this is not the case for the material obtained from experiments conducted for 3 min. As the material obtained at 3 min is the one in which star-shaped calcite dominates, it was chosen to be the one used for more advanced characterization studies. In the FTIR spectra presented in figure 5, it can be seen that there is no gaspeite ($NiCO_3$) in

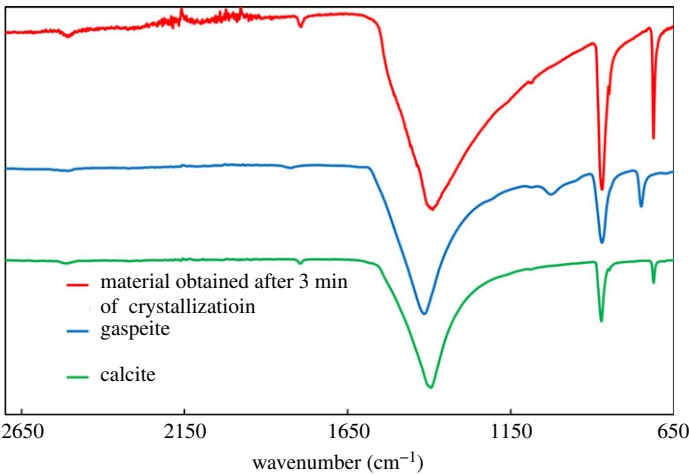

**Figure 5.** The FTIR spectrum of material obtained from the crystallization of 6 g of $CaCl_2 \cdot 2H_2O$ in the presence of 0.4 g of $NiCl_2 \cdot 6H_2O$ for 3 min, compared with the FTIR spectra of calcite and gaspeite. (Online version in colour.)

**Table 1.** The intensities (in arb. units) of the characteristic peaks of calcite and aragonite identified in samples obtained from the crystallization of 6 g of $CaCl_2 \cdot 2H_2O$ in the presence of 0.4 g of $NiCl_2 \cdot 6H_2O$ at different time periods and the ratios of the intensities of the characteristic peaks.

| | crystallization time | | | | |
|---|---|---|---|---|---|
| | 3 min | 5 min | 24 h | 48 h | 240 h |
| peak 1 (calcite, 29.55°) | 2345.43 | 996.46 | 829.78 | 1440.25 | 1712.85 |
| peak 2 (aragonite, 26.25°) | 117.18 | 176.19 | 741.05 | 727.23 | 900.89 |
| peak 2 (aragonite, 27.25°) | 172.63 | 536.19 | 431.97 | 451.64 | 584.02 |
| peak 1/peak 2 | 20.02 | 5.66 | 1.12 | 1.98 | 1.90 |
| peak 2/peak 3 | 13.59 | 1.86 | 1.92 | 3.19 | 2.93 |
| peak 2/peak 3 | 0.68 | 0.33 | 1.72 | 1.61 | 1.54 |

the samples of star-shaped calcite. The same conclusion cannot be easily drawn from examination of the corresponding XRD patterns, owing to the overlap between the calcite and the gaspeite peaks.

Figure 6 shows the vapour pressure of vaterite, aragonite, star-shaped calcite and rhomnbohedral calcite, as measured using TGA. In all the cases, the mass change, adjusted to the amount of physically bound water, was about 44%, corresponding to the formation of CaO. Vaterite, as the least stable $CaCO_3$ polymorph, has the highest vapour pressure, followed by aragonite. Rhombohedral calcite and star-shaped calcite have very similar vapour pressures. The slope of the lines of best fit in figure 6 gives the enthalpy of fusion for each case (the values can be seen in table 2).

Figure 7 summarizes the results from IGC measurements. In particular, figure 7a shows the $\gamma^{LW}$ and $\gamma^{AB}$ components of the surface energy. Figure 7b shows the breakdown of $\gamma^{AB}$ to $\gamma^{+}$ and $\gamma^{-}$. Finally, figure 7c depicts the total surface energy as well as the work of adhesion with water (at room temperature), calculated from equation (2.3).

The deconvolution of the surface energy maps gave the facet-specific surface energies of the main crystal facets exhibited in each sample. Through this *in silico* analysis the different components of the surface energy of the different facets as well as the relative surface area for

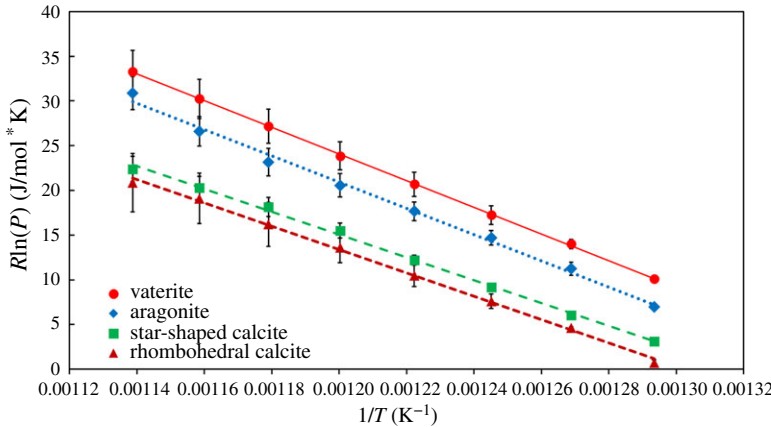

**Figure 6.** Arrhenius-type plot, showing the variation of the vapour pressure (*P*) with temperature (*T*). The slope of the lines corresponds to the enthalpy of fusion. (Online version in colour.)

each facet can be calculated. For the calculations, it was assumed that aragonite and star-shaped calcite exhibit two main and distinct crystal facets, whereas rhombohedral calcite was assumed to exhibit only one crystal facet, the (104). The selection of two distinct facets for the star-shaped calcite was driven by the fact that a combination of two distinct facets was found to give a better fitting behaviour. The discussion presented below is not significantly affected if a single distinct facet was chosen. Table 3 summarizes the facet-specific surface energies obtained, along with the estimated per cent coverage of each facet.

## 4. Discussion

The XRD patterns from figure 3 suggest that the crystallization proceeds in terms of the Ostwald rule of stages, with vaterite emerging first. Previous studies suggest that, in similar situations, the presence of transition metal cations, which can form trigonal carbonates, facilitates the nucleation of calcite [18]. Thus, calcite nucleates before aragonite (and immediately after vaterite), in the zone between the solubility lines of vaterite and aragonite. As calcite growth occurs in high supersaturation, it results in the formation of a high-surface-energy, star-shaped crystal habit. This has been demonstrated during the growth of barium sulfate; the formation of star-shaped crystals was attributed to the transition from a screw dislocation growth mechanism to a two-dimensional nucleation one [30]. Owing to its high surface energy, this crystal habit has higher solubility (according to equation (1.1)) than aragonite and hence it dissolves before aragonite crystals, which in turn dissolve before rhombohedral calcite crystals.

It is important to note that the two types of calcite share very similar vapour pressures. As vapour pressure is directly correlated to the Gibbs free energy of the solid, this finding means that, from a thermodynamic point of view, the two types of crystals are identical. This is also reflected in the corresponding enthalpies of melting. In general, as the two crystal habits have the same vapour pressure their bulk solid-state properties are identical. This is not the case for the surface properties: the surface energies of the two crystal habits of calcite vary significantly. The significant differences in the surface energy of different crystal habits result in differences in wettability and therefore in dissolution; this is a well-established concept in pharmaceutical technology [31]. Owing to their higher surface energy, especially their much higher $\gamma^{LW}$ and $\gamma^-$, the star-shaped crystals have about 30% higher affinity for work, as measured by the work of adhesion.

This is not unexpected. The crystal habit is determined by crystal growth conditions. Star-shaped crystals grow in much higher supersaturations than their rhombohedral counterparts. The crystallography of $CaCO_3$ enables the formation of crystal-shaped particles, as has already

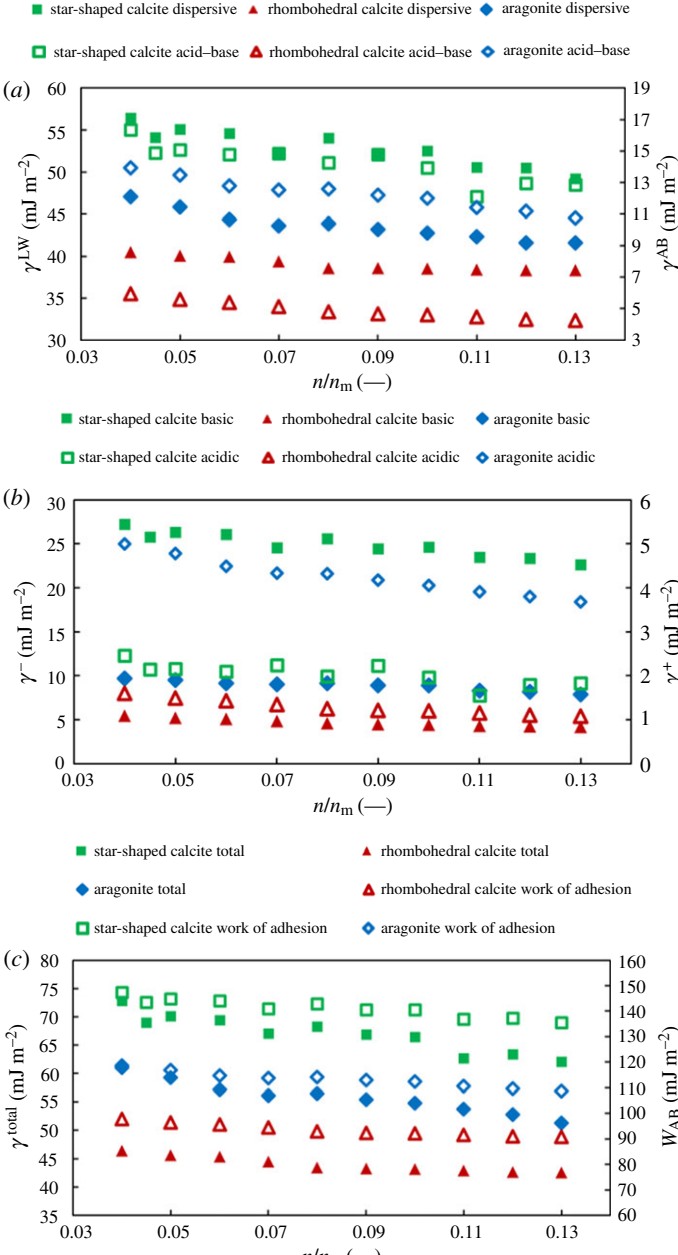

**Figure 7.** The surface energy maps of aragonite and star-shaped and rhombohedral calcite showing (*a*) the dispersive and the acid–base components of the materials, (*b*) the acidic and the basic components of the acid–base component and (*c*) the total surface energy and the work of adhesion with water. (Online version in colour.)

been demonstrated [32]. Under higher supersaturation, the thermodynamically most stable crystal habit, that is, the one minimizing the surface free energy of the crystal, is different from that in lower supersaturations. Based on the Ostwald–Freundlich equation [32,33] a higher supersaturation favours crystal habits with higher surface energy. In particular, this work shows that the star-shaped crystal habit is the thermodynamically favoured one under these conditions. It is important to highlight the presence of $Ni^{2+}$ ions, which promote the nucleation of calcite at

**Table 2.** The values of the enthalpy of fusion for the different materials investigated in this work.

| material | $-\Delta H_{\text{fusion}}$ (kJ mol$^{-1}$) |
|---|---|
| vaterite | 149.1 |
| aragonite | 146.7 |
| star-shaped calcite | 127.7 |
| rhombohedral calcite | 130.8 |

**Table 3.** The calculated values of the facet-specific surface energy for rhombohedral calcite, aragonite and star-shaped calcite crystals.

| | $\gamma^{\text{LW}}$ (mJ m$^{-2}$) | $\gamma^{+}$ (mJ m$^{-2}$) | $\gamma^{-}$ (mJ m$^{-2}$) | $\gamma^{\text{total}}$ (mJ m$^{-2}$) | coverage % |
|---|---|---|---|---|---|
| rhombohedral calcite | | | | | |
| facet 1 | 48 | 2 | 21 | 54.5 | 100 |
| aragonite | | | | | |
| facet 1 | 36 | 1.5 | 7 | 42 | 75 |
| facet 2 | 43 | 3.5 | 8 | 54.2 | 25 |
| star-shaped calcite | | | | | |
| facet 1 | 40 | 7 | 16 | 61.2 | 50 |
| facet 2 | 49 | 11 | 22 | 80.1 | 50 |

higher supersaturation [12], indicating the synergy between crystal nucleation and growth in the formation of metastable crystal habits.

According to the ideal solution theory, the solubility of a compound is calculated directly from the enthalpy of fusion/melting (the values are reported in table 2). Assuming that the heat capacity of melting is small, then the following equation can be formulated:

$$R\ln(X_i^{\text{ideal}}) = -\frac{\Delta H_m(T_m - T)}{(T_m T)}. \tag{4.1}$$

The melting temperatures for aragonite and calcite are known from the literature to be around 825°C and 1330°C, respectively [34]. By substituting the values from table 2 and the aforementioned melting temperatures in equation (4.1), the following solubility trend was obtained for a temperature near the boiling point:

$$X_{\text{aragonite}}^{\text{ideal}} > X_{\text{rhombohedral calcite}}^{\text{ideal}} > X_{\text{star-shaped calcite}}^{\text{ideal}},$$

where $\ln(X_{\text{aragonite}}^{\text{ideal}}) = -33.6$, $\ln(X_{\text{star-shaped calcite}}^{\text{ideal}}) = -33.7$ and $\ln(X_{\text{rhombohedral calcite}}^{\text{ideal}}) = -34.5$.

Using the $X^{\text{surface}}$ component of equation (1.1), the effects of facet-specific surface energies can be introduced for a given temperature and surface area of the particles involved in the process. A number of assumptions should be made when, first of all, calculating the value of $W_{il}$. It should be assumed that $W_{ll}$ is equal to the surface tension of water and that $W_{cc,i}$ is the facet-specific surface energy of the facet $i$. Furthermore, it should be assumed that the effects of temperature on the surface energy of the solid are small and that the dissolved solid does not change the surface tension of water dramatically. After incorporating these assumptions in equation (1.1), and assuming that the surface area of the particles is the same as that measured for octane, during the IGC experiments, the following trend for the values of $W_{il}$ can be calculated at $T = 298.15$ K

(the temperature at which we have reliable values for the surface energy components of water):

$$W_{il, \text{star-shaped calcite}} > W_{il, \text{aragonite}} > W_{il, \text{rhombohedral calcite}},$$

where $W_{il, \text{star-shaped calcite}} = 14.1 \, \text{J mol}^{-1}$, $W_{il, \text{aragonite}} = 6.57 \, \text{J mol}^{-1}$ and $W_{il, \text{rhombohedral calcite}} = 2.96 \, \text{J mol}^{-1}$.

This work of adhesion trend indicates that differences in interfacial energy, under certain conditions, can explain the swap in the stability sequence, in tandem with vapour pressure measurements. Under these circumstances, this swap has a direct impact on the sequence observed during the Ostwald rule of stages' transitions. Crystal habit is an important parameter that needs to be taken into account.

## 5. Conclusion

This work commences from an interesting case of the Ostwald rule of stages, suggesting that star-shaped calcite crystals may have higher solubility than aragonite crystals; this is a case of the interplay between polymorphism and surface properties. The vapour pressures of snowflake-shaped vaterite, needle-shaped aragonite and rhombohedral and star-shaped calcite crystals are determined. The surface energies of the same crystals, along with the facet-specific surface energies of the most important crystal facets, are determined for needle-shaped aragonite and rhombohedral and star-shaped calcite crystals.

The role of surface energy, in particular facet-specific surface energy, in the regular solution theory has been discussed in the past. However, this is probably the first example where this concept is addressed from a fundamental point of view. As the dissolution of crystals is of great importance in different industries, the findings of this work can have a substantial impact. Through the careful design of the crystal habit of drug substances, high-surface-energy crystals can be obtained to improve the solubility performance of drug products. The soft templating mechanism that resulted in the early nucleation of calcite, which is mentioned at the beginning of this work, can provide a new pathway in this direction.

Data accessibility. This article has no additional data.

Authors' contributions. E.H. did the experimental work and wrote the document. S.M.V. and J.D.L. provided mentorship and comments on the work. K.L.S.C. reviewed the document and provided directions during the experimental work.

Competing interests. We declare we have no competing interests.

Funding. The authors would like to acknowledge the funding and technical support from BP through the BP International Center for Advanced Materials (BP-ICAM), which made this research possible. This work was also partly funded by the EPSRC through the Prosperity partnership grant (EP/R00496X/1).

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
