## [Peer Review File · Proceedings. Mathematical, Physical, and Engineering Sciences]

Review History

RSPA-2021-0200.R0 (Original submission)

Review form: Referee 1

Is the manuscript an original and important contribution to its field?

Acceptable

Is the paper of sufficient general interest?

Good

Is the overall quality of the paper suitable?

Good

Can the paper be shortened without overall detriment to the main message?

Yes

Do you think some of the material would be more appropriate as an electronic appendix?

No

Do you have any ethical concerns with this paper?

No

Recommendation?

Major revision is needed (please make suggestions in comments)

Comments to the Author(s)

In this work, the authors have successfully made different shapes and polymorphs. It is interesting to report these different shapes and investigate the facet of these crystals. Then, the solubility was linked to the surface energy of different shapes and polymorphs, and the influence of different facets on solubility was discussed.

Some comments:

In the introduction, it is not clear if any other researchers have investigated the relation solubility with the surface energy in this system or other systems?

why Ni²⁺ was introduced?

any of these conditions repeated in experiments?

Figure 3 shows the obvious transition happening from 5 min to 24h? any date to show the transition progress?

Table 1, the peak values of XRPD are dependent on many factors? how authors minimize the influences?

Table 2, for facet 1 or 2, is that possible to mark them in the real crystal images?

Any polymorph change during the crystal growth or testing in TGA?

the work starts with an equation for dissolution, but the equation was not used in the discussion, will this equation help to understand the phenomena?

For other equations, such as Equ 4, will this equation help to make a quantity-comparison? rather than quality-comparison?

Review form: Referee 2

Is the manuscript an original and important contribution to its field?

Excellent

Is the paper of sufficient general interest?

Excellent

Is the overall quality of the paper suitable?

Excellent

Can the paper be shortened without overall detriment to the main message?

Yes

Do you think some of the material would be more appropriate as an electronic appendix?

No

Do you have any ethical concerns with this paper?

No

Recommendation?

Accept with minor revision (please list in comments)

Comments to the Author(s)

This is an interesting and detailed work on crystallization of calcium carbonate polymorphs in controlled conditions with design of morphology of calcite the stable morph to have high dissolution/ solubility in star shaped crystals. The applicability of this result is widespread and publication is recommended.

A few comments to improve/ correct the MS

1. Experimental - was de-aerated for 3 h with a N₂ stream (h is missing)
2. The additive NiCl₂.6H₂O which gives the desired morphology for stable calcite polymorph. A possible chemical binding or catalytic and templating effect may be added to make the rationale clear for readers how / why the morphology of calcite crystals changes from rhombohedral to star shaped.

Review form: Referee 3

Is the manuscript an original and important contribution to its field?

Excellent

Is the paper of sufficient general interest?

Excellent

Is the overall quality of the paper suitable?

Good

Can the paper be shortened without overall detriment to the main message?

Yes

Do you think some of the material would be more appropriate as an electronic appendix?

No

Do you have any ethical concerns with this paper?

No

Recommendation?

Accept with minor revision (please list in comments)

Comments to the Author(s)

This work discusses an interesting experimental study of vapor pressures and surface energies of a series of calcium carbonate polymorphs. The work highlights the importance of surface energy as a factor in polymorph solid state behaviour, and indeed how the design of high surface energy solids can infer improved solubility behaviour of crystal solid state material. A topic which should be of significant interest to the pharmaceutical world where the solubility of solid state dosage forms, especially hydrophobic molecules, is especially challenging, and this new insight could provide an interesting new strategy.

The vapor pressure study of the polymorphs is particularly interesting in Figure 6. The precise methodology is not mentioned, I assume it is Knudsen effusion. But I would like to see details included in the manuscript, including how the orifice size was determined as well as the sample temperature measurement if this was the method used. I think a Table reporting these heats of vaporisation should be included, as well as some comparison with literature values. The legend comment for Figure 6 "A plot similar to the previous one" does not seem very informative. The authors had included a detailed surface energy analysis of the crystal surfaces using IGC. This is a very comprehensive and informative data set. However, the methodology for deconvoluting this total surface energy data into facet specific surface energies is not explained at

all, beside the cryptic comment - in silico. Please provide a proper explanation of this important step in estimating the face specific data.

Decision letter (RSPA-2021-0200.R0)

26-May-2021

Dear Dr Hadjittofis,

On behalf of the Editor, I am pleased to inform you that your Manuscript RSPA-2021-0200 entitled "The role of surface energy on the apparent solubility of two different calcite crystal habits" has been accepted for publication subject to minor revisions in Proceedings A. Please find the referees' comments below.

The reviewer(s) have recommended publication, but also suggest some minor revisions to your manuscript. Therefore, I invite you to respond to the reviewer(s)' comments and revise your manuscript. Please note that we have a strict upper limit of 28 pages for each paper. Please endeavour to incorporate any revisions while keeping the paper within journal limits. Please note that page charges are made on all papers longer than 20 pages. If you cannot pay these charges you must reduce your paper to 20 pages before submitting your revision. Your paper has been ESTIMATED to be 15 pages. We cannot proceed with typesetting your paper without your agreement to meet page charges in full should the paper exceed 20 pages when typeset. If you have any questions, please do get in touch.

It is a condition of publication that you submit the revised version of your manuscript within 7 days. If you do not think you will be able to meet this date please let me know in advance of the due date.

To revise your manuscript, log into <https://mc.manuscriptcentral.com/prsa> and enter your Author Centre, where you will find your manuscript title listed under "Manuscripts with Decisions." Under "Actions," click on "Create a Revision." Your manuscript number has been appended to denote a revision.

You will be unable to make your revisions on the originally submitted version of the manuscript. Instead, revise your manuscript and upload a new version through your Author Centre.

When submitting your revised manuscript, you will be able to respond to the comments made by the referee(s) and upload a file "Response to Referees" in Step 1: "View and Respond to Decision Letter". You can use this to document any changes you make to the original manuscript. In order to expedite the processing of the revised manuscript, please be as specific as possible in your response to the referee(s).

IMPORTANT: Your original files are available to you when you upload your revised manuscript. Please delete any redundant files before completing the submission process.

When uploading your revised files, please make sure that you include the following as we cannot proceed without these:

- 1) A text file of the manuscript (doc, txt, rtf or tex), including the references, tables (including captions) and figure captions. Please remove any tracked changes from the text before submission. PDF files are not an accepted format for the "Main Document".

2) A separate electronic file of each figure (tif, eps or print-quality pdf preferred). The format should be produced directly from original creation package, or original software format.

3) Electronic Supplementary Material (ESM): all supplementary materials accompanying an accepted article will be treated as in their final form. Note that the Royal Society will not edit or typeset supplementary material and it will be hosted as provided. Please ensure that the supplementary material includes the paper details where possible (authors, article title, journal name). Supplementary files will be published alongside the paper on the journal website and posted on the online figshare repository (<https://figshare.com>). The heading and legend provided for each supplementary file during the submission process will be used to create the figshare page, so please ensure these are accurate and informative so that your files can be found in searches. Files on figshare will be made available approximately one week before the accompanying article so that the supplementary material can be attributed a unique DOI. Alternatively you may upload a zip folder containing all source files for your manuscript as described above with a PDF as your "Main Document". This should be the full paper as it appears when compiled from the individual files supplied in the zip folder.

Article Funder

Please ensure you fill in the Article Funder question on page 2 to ensure the correct data is collected for FundRef (<http://www.crossref.org/fundref/>).

Media summary

Please ensure you include a short non-technical summary (up to 100 words) of the key findings/importance of your paper. This will be used for to promote your work and marketing purposes (e.g. press releases). The summary should be prepared using the following guidelines:

*Write simple English: this is intended for the general public. Please explain any essential technical terms in a short and simple manner.

*Describe (a) the study (b) its key findings and (c) its implications.

*State why this work is newsworthy, be concise and do not overstate (true 'breakthroughs' are a rarity).

*Ensure that you include valid contact details for the lead author (institutional address, email address, telephone number).

Cover images

We welcome submissions of images for possible use on the cover of Proceedings A. Images should be square in dimension and please ensure that you obtain all relevant copyright permissions before submitting the image to us. If you would like to submit an image for consideration please send your image to proceedingsa@royalsociety.org

Open Access

You are invited to opt for open access, our author pays publishing model. Payment of open access fees will enable your article to be made freely available via the Royal Society website as soon as it is ready for publication. For more information about open access please visit <https://royalsociety.org/journals/authors/open-access/>. The open access fee for this journal is £1700/\$2380/€2040 per article. VAT will be charged where applicable. Please note that if the corresponding author is at an institution that is part of a Read and Publishing deal you are required to select this option. See <https://royalsociety.org/journals/librarians/purchasing/read-and-publish/read-publish-agreements/> for further details.

Once again, thank you for submitting your manuscript to Proceedings A and I look forward to receiving your revision. If you have any questions at all, please do not hesitate to get in touch.

Best wishes
 Raminder Shergill
 proceedingsa@royalsociety.org
 Proceedings A

on behalf of
 Dr Andy Sutherland
 Board Member
 Proceedings A

Reviewer(s)' Comments to Author:

Referee: 1

Comments to the Author(s)

In this work, the authors have successfully made different shapes and polymorphs. It is interesting to report these different shapes and investigate the facet of these crystals. Then, the solubility was linked to the surface energy of different shapes and polymorphs, and the influence of different facets on solubility was discussed.

Some comments:

In the introduction, it is not clear if any other researchers have investigated the relation solubility with the surface energy in this system or other systems?

why Ni²⁺ was introduced?

any of these conditions repeated in experiments?

Figure 3 shows the obvious transition happening from 5 min to 24h? any data to show the transition progress?

Table 1, the peak values of XRPD are dependent on many factors? how authors minimize the influences?

Table 2, for facet 1 or 2, is that possible to mark them in the real crystal images?

Any polymorph change during the crystal growth or testing in TGA?

the work starts with an equation for dissolution, but the equation was not used in the discussion, will this equation help to understand the phenomena?

For other equations, such as Equ 4, will this equation help to make a quantity-comparison? rather than quality-comparison?

Referee: 2

Comments to the Author(s)

This is an interesting and detailed work on crystallization of calcium carbonate polymorphs in controlled conditions with design of morphology of calcite the stable morph to have high dissolution/ solubility in star shaped crystals. The applicability of this result is widespread and publication is recommended.

A few comments to improve/ correct the MS

1. Experimental - was de-aerated for 3 h with a N₂ stream (h is missing)
2. The additive NiCl₂·6H₂O which gives the desired morphology for stable calcite polymorph. A possible chemical binding or catalytic and templating effect may be added to make the rationale clear for readers how / why the morphology of calcite crystals changes from rhombohedral to star shaped.

Referee: 3

Comments to the Author(s)

This work discusses an interesting experimental study of vapor pressures and surface energies of a series of calcium carbonate polymorphs. The work highlights the importance of surface energy as a factor in polymorph solid state behaviour, and indeed how the design of high surface energy solids can infer improved solubility behaviour of crystal solid state material. A topic which should be of significant interest to the pharmaceutical world where the solubility of solid state dosage forms, especially hydrophobic molecules, is especially challenging, and this new insight could provide an interesting new strategy.

The vapor pressure study of the polymorphs is particularly interesting in Figure 6. The precise methodology is not mentioned, I assume it is Knudsen effusion. But I would like to see details included in the manuscript, including how the orifice size was determined as well as the sample temperature measurement if this was the method used. I think a Table reporting these heats of vaporisation should be included, as well as some comparison with literature values. The legend comment for Figure 6 "A plot similar to the previous one" does not seem very informative.

The authors had included a detailed surface energy analysis of the crystal surfaces using IGC. This is a very comprehensive and informative data set. However, the methodology for deconvoluting this total surface energy data into facet specific surface energies is not explained at all, beside the cryptic comment - in silico. Please provide a proper explanation of this important step in estimating the face specific data.

Author's Response to Decision Letter for (RSPA-2021-0200.R0)

See Appendix A.

Decision letter (RSPA-2021-0200.R1)

22-Jul-2021

Dear Dr Hadjittofis

I am pleased to inform you that your manuscript entitled "The role of surface energy on the apparent solubility of two different calcite crystal habits" has been accepted in its final form for publication in Proceedings A.

Our Production Office will be in contact with you in due course. You can expect to receive a proof of your article soon. Please contact the office to let us know if you are likely to be away from e-mail in the near future. If you do not notify us and comments are not received within 5 days of sending the proof, we may publish the paper as it stands.

As a reminder, you have provided the following 'Data accessibility statement' (if applicable). Please remember to make any data sets live prior to publication, and update any links as needed when you receive a proof to check. It is good practice to also add data sets to your reference list.
Statement (if applicable):

Open access

You are invited to opt for open access, our author pays publishing model. Payment of open access fees will enable your article to be made freely available via the Royal Society website as soon as it is ready for publication. For more information about open access please visit

<https://royalsociety.org/journals/authors/which-journal/open-access/>. The open access fee for this journal is £1700/\$2380/€2040 per article. VAT will be charged where applicable.

Note that if you have opted for open access then payment will be required before the article is published – payment instructions will follow shortly.

If you wish to opt for open access then please inform the editorial office (proceedingsa@royalsociety.org) as soon as possible.

Your article has been estimated as being 17 pages long. Our Production Office will inform you of the exact length at the proof stage.

Proceedings A levies charges for articles which exceed 20 printed pages. (based upon approximately 540 words or 2 figures per page). Articles exceeding this limit will incur page charges of £150 per page or part page, plus VAT (where applicable).

Under the terms of our licence to publish you may post the author generated postprint (ie. your accepted version not the final typeset version) of your manuscript at any time and this can be made freely available. Postprints can be deposited on a personal or institutional website, or a recognised server/repository. Please note however, that the reporting of postprints is subject to a media embargo, and that the status the manuscript should be made clear. Upon publication of the definitive version on the publisher's site, full details and a link should be added.

You can cite the article in advance of publication using its DOI. The DOI will take the form: 10.1098/rspa.XXXX.YYYY, where XXXX and YYYY are the last 8 digits of your manuscript number (eg. if your manuscript number is RSPA-2017-1234 the DOI would be 10.1098/rspa.2017.1234).

For tips on promoting your accepted paper see our blog post: <https://royalsociety.org/blog/2020/07/promoting-your-latest-paper-and-tracking-your-results/>

On behalf of the Editor of Proceedings A, we look forward to your continued contributions to the Journal.

Sincerely,
Raminder Shergill
proceedingsa@royalsociety.org

on behalf of
Dr Andy Sutherland
Board Member
Proceedings A

Appendix A

The authors thank the Review panel for its fruitful and thought-provoking comments. A document highlighting the main modifications in the document is provided to facilitate the review process.

On the comments by the Reviewer 1:

The authors thank the Reviewer for her meticulous study of the manuscript and her comments. Reviewer 1 summarised in a few lines the key message we were trying to convey.

In the introduction, it is not clear if any other researchers have investigated the relation solubility with the surface energy in this system or other systems?

The authors thank the reviewer for the comment. It is true that the introduction is the part of the document where the authors need to engage the audience, therefore, this comment allows us to improve the quality of our work.

The impact of surface energy anisotropy in apparent solubility has been introduced by Gibbs. It found some application in the area of mineralogy. Later on, a number of researchers have engaged into this field, from different angles, but none has gone to the level of detail we did, with the combination of vapour pressure and surface energy anisotropy.

The authors believe that the lack of a methodology allowing for the determination of the surface energy anisotropy of statistically significant samples of particles, hindered the development of this field of research. The in silico analysis of IGC results by means described in this work overcomes this challenge.

why Ni²⁺ was introduced?

The effect of calcite's crystal habit, on the mineral's solubility, is in the epicentre of this work. Different strategies have been developed to leverage the crystal habit of calcite and other crystals. The addition of Ni²⁺ provides great leverage over the crystal habit, as, at very high supersaturations (> 5000), it enables the growth of the star-shaped calcite crystals in a very short time ($t < 10$ mins). These crystals, if not isolated, they will start dissolving owing to the Ostwald rule of stages, in the expense of aragonite. The exact mechanism via which certain cationic species, including Ni²⁺, yield star-shaped calcite crystals is still a matter of investigation. Only phenomenologically we know it works. The small [Ni²⁺] does not allow for the formation of gaspeite. Furthermore, the vapour pressure measurements indicate that the possibility of inclusions is highly unlikely. If inclusions were formed, the impact on the crystal lattice would have been manifested in an increase in vapour pressure.

any of these conditions repeated in experiments?

Reproducibility is important, especially during crystallisation at high supersaturations. The conditions, used for the isolation of the different types of crystals, were selected after a phenomenological investigation of the crystallisation of CaCO₃ under sweet and anoxic conditions and in the presence of different cationic additives. The experiments were repeated, and reproducibility was verified. The characterisation was performed on a single batch of material.

Figure 3 shows the obvious translation happening from 5 min to 24h? any data to show the transaction progress?

Detailed SEM imaging suggest that the star-shaped calcite crystals shrink with increasing time. Similarly, needle shaped aragonite crystals dissolve as well. On the other hand, rhombohedral calcite crystals appear later on and they grow. This pattern combined with the XRD results suggest that the crystallization is determined by the Ostwald rule of stages. This is in-line with the theory suggesting that the Ostwald rule is triggered at relatively high supersaturations. It is not clear yet, whether star-shaped calcite crystals shrink to a certain minimum size and then they regrow as rhombohedral crystals.

Table 1, the peak values of XRPD are dependent on many factors? how authors minimize the influences?

It is true that the peak quality can be influenced by numerous factors. The presence of amorphous and/or nanoparticles is one of those. The SEM investigation did not reveal the presence of such particles. A large amount of sample was used, in a back-loaded sample holder. As the material was relatively free flowing and fine, no mechanical force was exerted on it, to assist in the packing. Thanks to the fine nature of the particles, the surface of the sample can be considered flat with respect to the sample holder. For the crystals produced at 3 and 5 mins, in the presence of Ni²⁺, the material was unloaded from the sample holder, mixed with the mother batch and then a new sample was taken for a second measurement. The same procedure was repeated once again. The three measurements were identical. This point was also amended in the document.

Table 2, for facet 1 or 2, is that possible to mark them in the real crystal images?

Crystals are faceted entities, with each facet corresponding to a different crystallographic plane. The authors were not able to grow a macroscopic single crystal of star-shaped calcite, to perform the appropriate studies required to confidently provide the requested drawing. The authors assumed two crystal facets as this number was allowing better fitting of experimental and simulated results. The use of a single major crystallographic plane was not going to change the outcomes of the discussion drastically.

Any polymorph change during the crystal growth or testing in TGA?

This is an excellent point that the authors have considered; albeit it was not mentioned in the text for brevity. The enantiotropic relationship, between calcite and aragonite, is a very important research question. Therefore, it may be the case that the TGA measurement may be impacted by a potential in situ transformation of calcite to aragonite. During the TGA experiments, the material was exposed at a temperature gradient in an open pan connected to a mass spectrometre. The existing phase diagrams show that, for the range of temperatures used, the pressures required for an enantiotropic transition are orders of magnitude higher than those encountered in the TGA set-up. Therefore, the possibility of enantiotropic transformation is minimum.

- (1) Zimmermann, H.D. Equilibrium Conditions of the Calcite/Aragonite Reaction between 150°C and 350°C *Nature Physical Science*, **1971**, 231, 203–204

the work starts with an equation for dissolution, but the equation was not used in the discussion, will this equation help to understand the phenomena?

This equation (Eq. 1) provides the mathematical framework to understand why the difference in surface energy anisotropy can alter the apparent solubility sequence. To address this, the authors amended the document to compare the results of Eq. 1 with those of Eq. 4 (Eq. 5 in the revised manuscript) quantitatively.

For other equations, such as Eq 4, will this equation help to make a quantity-comparison? rather than quality-comparison?

A more quantitative approach will substantially elevate the quality of the manuscript. An appropriate amendment was done to address this.

On the comments by the Reviewer 2

The authors thank the Reviewer for its fruitful and thought-provoking comments. Its comment is very helpful and gives us confidence that our work can have a greater impact industrially

1. Experimental - was de-aerated for 3 h with a N2 stream (h is missing)

This work is part of a greater effort to investigate phenomena in sweet and anoxic conditions. Such conditions are encountered industrially in the oil and gas industry, but they are also relevant to other scientific disciplines. In particular, disciplines investigating phenomena in Earth's primordial period are often considering sweet and anoxic conditions. Therefore, it was important to remove all the oxygen, to ensure consistency with the broader context of this work. Furthermore, the removal of oxygen eliminates the potential formation of oxides, reducing the complexities in terms of results' analysis.

A relatively large flow rate of gas was used (200 mL/min). The bubbling was allowing for better mass transfer. The time was selected, considering that, in literature, for larger volumes de-aeration with lower nitrogen flow rate was assumed to yield sufficient removal of oxygen.

The document was amended to accommodate this point.

(2) Sadeek, S.A.; Williams, D.R.; Campbell K.L.S. Using sodium thiosulphate for carbon steel corrosion protection against monoethanolamine and methyldiethanolamine *Int. J. Greenh. Gas Control.*, **2018**, 74, 206-218

2. The additive NiCl₂·6H₂O which gives the desired morphology for stable calcite polymorph. A possible chemical binding or catalytic and templating effect may be added to make the rationale clear for readers how / why the morphology of calcite crystals changes from rhombohedral to star shaped.

The reviewer raises a very important point, going to the epicentre of our effort to elucidate the effect of certain ions in crystallization at high supersaturations. Experiments with plethora of cationic species allow

us to identify the potential of new pathways of action, beyond the established. A potential templating effect is one of the mechanisms proposed to explain our observations. This is work in progress which will be greatly benefited from the publication of this manuscript and further characterisation studies of samples prepared in the presence of different cations.

On the comments by the Reviewer 3:

The authors thank the Reviewer for its encouraging comments, as well as the careful consideration of the manuscript, revealed from its summary

Comments to the Author(s):
The vapor pressure study of the polymorphs is particularly interesting in Figure 6. The precise methodology is not mentioned, I assume it is Knudsen effusion. But I would like to see details included in the manuscript, including how the orifice size was determined as well as the sample temperature measurement if this was the method used. I think a Table reporting these heats of vaporisation should be included, as well as some comparison with literature values. The legend comment for Figure 6 "A plot similar to the previous one" does not seem very informative.

The measurement of the vapour pressure is at the epicentre of this investigation. Therefore, the authors fully understand that a more elaborated work should have been done to describe the methodology used. The methodology used is similar as in a Knudsen cell. The foundation of this methodology is the following equation, for the rate of mass loss (dm/dt), per unit area (α), during the heating of a compound:

$$\left(\frac{1}{\alpha}\right) \frac{dm}{dt} = p\alpha \sqrt{\frac{M}{2\pi RT}}$$

In the above equation, p is the vapour pressure, α is a vapourisation coefficient, M is the molar mass of the vapour, R and T are the universal gas constant and temperature respectively.

The measurements were performed in a TGA equipment (Netzsch STA449), coupled with a mass spectrometre. The value of α is determined by the diameter of the orifice, which in this case is assumed to be equal to the diameter of the TGA's crucible ca. 5 mm. The vapourisation constant, for the particular set-up, was determined using benzoic acid as the calibration standard. Benzoic acid has a very well reported vapour pressure curve. Thus, it can be used to determine the value of the vapourisation coefficient at the particular arrangement. The document was amended to accommodate this comment.

The caption of the Figure was corrected according to the Reviewer's suggestion.

The authors had included a detailed surface energy analysis of the crystal surfaces using IGC. This is a very comprehensive and informative data set. However, the methodology for deconvoluting this total surface energy data into facet specific surface energies is not explained at all, beside the cryptic comment - in silico. Please provide a proper explanation of this important step in estimating the face specific data.

The authors recognise the need to expand on the *in-silico* approach used to obtain facet specific surface energy from IGC measurements. In this direction, an amendment was done in the document. Crystalline particles are faceted entities with each facet corresponding to a different crystallographic plane. The facets are named by means of Miller indexing. Each crystallographic facet has different surface energy,

the value of which is in great extent determined from its surface chemistry. The abundance of each of the facets on a particular particle determines the surface energy of the particle.

For instance, rhombohedral calcite crystals, mainly, express a single crystallographic facet, the (104) one. A needle shaped crystal, like aragonite is assumed to express two main crystallographic facets, those running along its length; the surface area of the facets at the two ends of the needle is considered very small to be considered. Finally, the star-shaped calcite crystal, was, assumed to express two main crystallographic facets. The authors assumed two crystal facets as this number was allowing better fitting of experimental and simulated results.

IGC allows for the measurement of the surface energy, using different solvent probes. At small probe concentrations only the highest surface energy facets are considered. As the probe concentration increases, facets with lower surface energy come into play. Thus, a typical IGC plot starts at relatively high values and then, as the surface coverage increases, it reaches a plateau corresponding to an “average” value of the surface energy of the sample.

The authors have simulated the adsorption process taking place during an IGC measurement. Using an optimisation algorithm they were able to predict the values of the facet specific surface energy and the relative amount of the surface of each facet as a fraction of the total surface area, corresponding to the materials under investigation.